# Nicarbazin Residue in Tissues from Broilers Reared on Reused Litter Conditions

**DOI:** 10.3390/ani12223107

**Published:** 2022-11-10

**Authors:** Vivian Feddern, Gerson N. Scheuermann, Arlei Coldebella, Vanessa Gressler, Gizelle C. Bedendo, Luizinho Caron, Antonio C. Pedroso, Danniele M. Bacila, Anildo Cunha

**Affiliations:** 1Embrapa Suínos e Aves [Embrapa Swine and Poultry], BR 153, km 110, Concórdia 89715-899, SC, Brazil; 2Embrapa Solos [Embrapa Soil], Setor de Transferência de Tecnologia, Jardim Botânico 1024, Rio de Janeiro 22460-000, RJ, Brazil; 3Instituto Federal Catarinense, Concórdia 89703-720, SC, Brazil; 4Departamento de Engenharia Química, Universidade Federal do Paraná, Curitiba 80060-000, PR, Brazil

**Keywords:** chicken breast fillet, chicken liver, chemical residues, DNC residues, liquid chromatograph, poultry farming

## Abstract

**Simple Summary:**

The poultry production worldwide needs to use feed additives in order to avoid disease. One of the most common diseases is coccidiosis, which affects the poultry gut. In case the disease happens, birds stop eating, which might cause their death as well as generating great losses to the farmer. Nicarbazin is one of the widely used anticoccidials to prevent coccidiosis. However, some non-compliances have been reported by the regulatory agencies, which made us explore chicken litter to verify if it was a contamination source in chicken tissues because birds have coprophagic habits. Therefore, our team conducted an experiment with three different nicarbazin treatments commonly used in commercial poultry production, under reused litter conditions. We evaluated the possible residue left from this anticoccidial in fillet and liver during 2 years. The outcomes have relevant implications: (a) litter is not a cause of contamination, allowing farmers to reuse it for economical purposes; and (b) chicken tissues are safe for human consumption.

**Abstract:**

Nicarbazin (NCZ) is a worldwide used anticoccidial in poultry farming to avoid coccidiosis disease when chickens are reared on conventional poultry litter. If proper dosage and withdrawal time are not followed, the component dinitrocarbanilide (DNC) of NCZ may be present in chicken tissues, therefore posing a risk to consumers if the residues are above 200 µg/kg. Litter reuse is a common and important practice in commercial chicken production. Literature is lacking about the influence of litter reuse on DNC deposition in chicken tissues and its impact on food safety. We aimed to evaluate DNC residues in breast and liver by LC-MS/MS from broilers from an experiment with 10 consecutive flocks during 2 years. The experiment included three treatments containing NCZ in the diet (T1 = 125 mg/kg, 1–21 d; T2 = 125 mg/kg, 1–32 d; T3 = 40 mg/kg, 1–32 d). DNC residues in chicken breast at 21 d in T1 ranged from 648.8–926 µg/kg, at 32 d in T2 and T3 varied, respectively, from 232–667 µg/kg and 52–189 µg/kg. Regarding liver, DNC residues at 21 days in T1 ranged from 11,754–15,281 µg/kg, at 32 days in T2 and T3 varied, respectively, from 10,168–15,021 µg/kg and 2899–4573 µg/kg. When NCZ was withdrawn from feed, DNC residues dropped to <LOQ at 42 d in all treatments. Therefore, the reuse of poultry litter does not compromise food safety regarding DNC residues in chicken tissues, as shown herein up to 10 flocks.

## 1. Introduction

Coccidiosis is an enteric disease that regularly affects broiler chickens. When not controlled, this disease caused by protozoa from the *Eimeria* genus has severe implications for poultry production [1,2]. As a preventive measure against coccidiosis outbreaks, broiler feed often contains nicarbazin (NCZ), an additive with significant anti-*Eimeria* action [3,4]. However, this approach has been a matter of concern because NCZ active component (4,4-dinitrocarbanilide or DNC) can occur in chicken meat if proper dosage and withdrawal period are not followed [5,6,7].

The major worldwide chicken producers (Brazil, USA, China, and EU) approve NCZ as an anticoccidial for broilers under specific rules. In Brazil, for instance, regulations allow its addition to feed limited to 125 mg/kg and require a 10-day NCZ-free diet before slaughter. Other countries are less restrictive, allowing a withdrawal of 4–5 days [8] or even 1 day [9]. The withdrawal period is crucial for depletion of DNC incurred in chicken tissues to levels below the maximum residue limit (MRL) of 200 μg/kg as established by *Codex Alimentarius* international standards [10]. Nevertheless, Brazilian surveillance programs still identify chicken samples with DNC concentration exceeding the mentioned MRL [11]. The appearance of this residue may be related to unintended factors, such as deep-litter reuse, a common practice in poultry farming. 

Broilers receiving NCZ additive in the diet eliminate unchanged DNC through their excreta. NCZ-rich litter along the coprophagic habit of broilers may result in occasional DNC intake during the withdrawal period when providing anticoccidial-free rations. Earlier studies have indicated the possibility of DNC recirculating from litter into the chicken tissues [12,13,14,15,16]. Consequently, there are concerns that prolonged litter reuse along successive flocks, when NCZ is fed to broilers, increases the chances of finding DNC in edible tissues above the regulatory limit. However, this risk under commercial poultry production has never been addressed. Therefore, this study aimed to assess whether the successive use of litter in NCZ-feeding programs implies in chicken meat contamination with DNC, at a level higher than MRL.

## 2. Materials and Methods

### 2.1. Care and Use of Animals

This research was conducted in accordance with the ethical guidelines of the National Council for the Control of Animal Experimentation [17]. Furthermore, the Ethics Committee on Animal Use (protocol no. 013/2016) and the Biosecurity Committee (protocol no. 009/2018), both from Embrapa Suínos e Aves (Embrapa Swine and Poultry), approved all experimental procedures.

### 2.2. Litter Reuse Experiment

The research was conducted between October 2018 and June 2020 in experimental facilities that resemble poultry commercial field conditions. Ten successive flocks of broilers (42-day growth cycles) were raised on a single poultry litter, according to a randomized block design. In each flock, 864 1-day-old broiler chicks were housed in 24 collective pens (18 females and 18 males per pen, ~12 chicks/m^2^), as shown in Figure 1. 

Three treatment diets were sorted by chance and assigned to the pens within eight replicates at different locations in the broiler house. Basal diets (Appendix A) were formulated to meet the nutritional requirements of broilers at three growing stages (starter, 1–21 days; grower, 22–32 days; and finisher, 33–42 days) [18]. The experimental treatments, based on anticoccidial programs used regularly for poultry, consisted of different supplementary additive NCZ conditions over the basal diet as follows:T1 = NCZ (125 mg/kg) fed from 1–21 days; NCZ-free feed from 22–42 days (withdrawal);T2 = NCZ (125 mg/kg) fed from 1–32 days; NCZ-free feed from 33–42 days (withdrawal);T3 = NCZ (40 mg/kg) + maduramicin (3.75 mg/kg) fed from 1–32 days; NCZ-free feed from 33–42 days (withdrawal).

The first flock was bedded on a 5–7 cm layer of clean wood shavings in all pens [19]. Throughout the 10 flocks, the assignment treatments of each pen remained unchanged. The same litter was kept (reused) in the pens for the nine subsequent flocks.

A 2-week downtime was established between grow-outs (flocks) to perform the following procedures: (a) flamethrower application on litter surface to burn the feathers; (b) litter de-caking in compacted or crusted areas; (c) litter turning accompanied by quicklime incorporation; and (d) clean wood shavings (~1–2 cm) spreading over the brooders inside the pens. Feed and water were provided *ad libitum* throughout the experiment. All feeders were properly emptied between growth stages within each flock. 

### 2.3. Pre-Slaughter Fasting Trial

On the last day of the litter reuse experiment (as soon as flock 10 was finished), 1 female and 1 male were removed from each of the pens assigned to treatment 2. This group of 16 broilers (42-day-old) was housed on reused litter in a single collective pen of this same treatment 2, where they remained without access to feed. Only water was provided *ad libitum*. After 6 h or 12 h fasting, 8 broilers (4 females and 4 males) were transferred to transport cages, sent for slaughter and collection of breast fillet and liver as detailed in the next Sampling section.

### 2.4. Sample Collection

All experimental feeds were sampled to check NCZ content (determined as DNC) or its absence. Collections of broiler tissues and poultry litter were performed in the equidistant flocks 1, 4, 7, and 10. For sampling, one female and one male (corresponding to the gender average weight within the experimental unit) were removed from each pen at 21, 32, and 42 days old. After fasting (3–4 h) inside transport cages, the 48 broilers *per* period were slaughtered (cervical dislocation followed by bleeding during 3 min) for immediate collection of skinless breast fillet (*pectoralis major* muscle) and liver. The samples were individually packed in plastic bags and kept at 4 °C for further processing within 24 h. In addition, one pooled litter sample *per* pen was taken on days 1, 21, 32, and 42 at the particular flocks. Litter samples were packed in plastic bags and processed within 24 h.

### 2.5. Sample Processing

Feed. As received (300 g), samples were entirely processed in a Tecator Cemotec 1090 mill (Hillerød, Denmark), packed in plastic bags, and stored at −20 °C until DNC determination.

Breast and liver. Breast fillets (entire sample cut into small cubes) and whole livers were individually weighed into aluminum trays. The samples were frozen at −20 °C for at least 24 h and then lyophilized for 48 h in an LJI-030 freeze-dryer (JJCientífica, São Carlos, SP, Brazil). After weighing, the freeze-dried samples were ground in an IKA A11 Basic analytical mill (Staufen, Baden-Württemberg, Germany), packed in plastic bags, and stored at −20 °C until DNC determination. Moisture loss was calculated using sample mass before and after the freeze-drying process.

Poultry Litter. Samples (200–300 g) were weighed into aluminum trays and left to air dry at 20 °C for 48 h. The dried samples were processed in a Tecator Knifetec 1095 mill (Hillerød, Denmark), which were packed in plastic bags and stored at −20 °C until DNC determination.

### 2.6. DNC Determination 

The procedures for analyzing DNC in poultry feed [20], chicken breast and liver [21], and litter are described in Appendix A, including reagents, solutions, extraction steps, and chromatographic conditions for HPLC-UV or LC-MS/MS. This file also contains details on NCZ content in the experimental feeds (Appendix A) and accuracy and precision of the applied methods (Appendix A).

### 2.7. Statistical Analysis

Data of DNC concentration in the litter, breast fillet, and liver were subjected to analysis by mixed models for repeated measures (MMRM), considering the effects of the block, treatment, flock, age, and the interactions between these last three causes of variation. The PROC MIXED procedure of SAS [22] was used to fit and evaluate three types of variance-covariance matrix structures, contemplating the dependencies between flocks and ages [23]. The structure with the lowest value of Akaike Information Criterion (AIC) was selected for use in further analyses. The restricted maximum likelihood (REML) approach was applied as estimation method. Whenever detected by the *F*-test, significant effects (*p* ≤ 0.05) were unfolded using the protected *t*-test to compare means, two by two.

## 3. Results

We monitored DNC residues left by NCZ in breast fillet and liver from broilers reared on the same litter in different flocks (1, 4, 7 and 10). The main finding corresponds to data reported at 42 days of age in which no detectable DNC concentration was found in chicken breast fillet and liver. 

According to descriptive levels of probability (*F*-test) for repeated measures, DNC concentration in target tissues and litter was affected (*p* ≤ 0.05) by treatment, age, and flock. For data detailing, the subsequent analysis considered the effect of interaction between these causes of variation (treatment x age x flock), as demonstrated in Appendix A.

DNC accumulated in breast fillet and liver of broilers from 10 consecutive flocks reared on the same litter are shown, respectively, in Table 1 and Appendix A (see Appendix A). Clearly, higher concentrations were observed in the liver as compared to the values in breast fillet. Despite responding to the NCZ dosage, deposition of DNC followed the same predictable profile in both tissues regardless of treatment or flock. The concentration far exceeded the *Codex Alimentarius* MRL (200 µg/kg), while broilers received the additive for breast fillet but achieved a non-quantifiable amount afterwards. Similar behavior was observed for the liver but with low quantifiable amounts after the diet withdrawal period.

Figure 2 shows results of DNC deposition in chicken liver at 42 days, indicating compliance (<MRL) in all treatments. The highest DNC concentration (122 µg/kg) was found in T2, treatment in which the birds consumed the highest NCZ amount for the longest time.

Age effect (*p* ≤ 0.05) was observed on DNC incurred in both target tissues in most treatment-flock interactions (Table 1 and Appendix A). In treatment T1, higher concentration levels were observed until 21 days (the first sampling age and the last NCZ feeding day for this treatment). T2 and T3 showed the same behavior but prolonged to 32 days of age (the last time NCZ was fed). Lower concentration levels were observed for T3, because less NCZ was fed in this treatment. As a consequence of the withdrawal period, DNC concentration depleted until traces below the limit of quantification (LOQ = 20 µg/kg) for breast fillet at 42 days for all treatments, and earlier at 32 days for T1. In the liver, DNC residue dropped to a safe concentration (<MRL = 200 µg/kg, according to *Codex Alimentarius*) as well, albeit occurring at a quantifiable level in most situations.

Flock effect (*p* ≤ 0.05) exists for both tissues (Table 1 and Appendix A) being uneven among ages. At 32 days, DNC residue differed significantly between flocks in the both tissues at T2 but only in the breast at T3. The data did not show a tendency to increase DNC deposition along the flocks and no difference (*p* > 0.05) between flocks was observed in either breast or liver at 42 days.

By plotting histograms (not shown) of the individual results at 42 days from all treatments, cases of DNC above the MRL appeared randomly only among the livers (4 in flock 7; and 1 in each of the other flocks), representing 3.7% within this sample group (n = 192). Over the 10 successive flocks subjected to NCZ feeding, we observed no systematic increase in non-compliant samples for DNC contamination (breast fillet or liver) associated with exposure of broilers to reused litter.

Figure 3 and Appendix A present the DNC content in litter used to house multiple flocks of broilers consuming NCZ. Regarding flock 1 in particular, the highest value within treatments were confirmed on the latter day of NCZ administration (day 21 in T1, and 32 in T2 and T3), falling by approximately half after finishing at 42 days. DNC concentration varied significantly (*p* ≤ 0.05) with age in each treatment throughout flock 1. Flocks 1, 7 and 10 in T1 (Figure 3a) had the same descending profile, with a significant peak on the last day of additive supply (21 days). In Flock 1 and 10, there was age effect (*p* ≤ 0.05). Significant (*p* ≤ 0.05) variations over age occurred at T2 in flocks 1, 4 and 10 (Figure 3b). For T3, the comparison showed no differences (*p* > 0.05) between ages in flocks 4, 7 e 10 (Figure 3c).

Between flocks, at T1 and T2, DNC content in poultry litter showed highly significant (*p* ≤ 0.05) variations. The highest DNC concentrations in these treatments occurred in flock 4. Overall, the content at flock 7 fell back to the first flock level, returning to slightly higher values in flock 10. For T3, the DNC concentration was below the other treatments with significant flock effect (*p* ≤ 0.05) at 21, 32 and 42 days.

## 4. Discussion

Litter reuse has become a helpful economic and microbiologically safe practice for a sustainable poultry farming [24]. However, its possible involvement with the deposition of anticoccidial residues in chicken tissues remains unclear. Attempts to clarify the facts surrounding this issue gain importance in the actual growing food safety requirements.

Our study suggests that poultry litter reuse in different flocks of NCZ-fed broilers does not represent a cause for meat contamination with DNC at unsafe levels. These findings come from data obtained at 42 days (broiler ready for consumption) and make sense based on the following reasons: (a)Low litter consumption during the withdrawal period, which restricts the DNC amount recirculating in chickens. Indeed, litter (wood chips or sawdust) accounts for up to 3% of what a broiler consumes from 1 to 21 days post-hatching but represents less than 1% between 22–49 days of age [25];(b)Only a minimal DNC fraction that returns through litter is readily available for intestinal absorption, making unlikely the possibility of edible tissues contamination at an unsafe level. This claim arises from the low bioavailability of DNC, once it is characterized as a poorly water-soluble compound (2 µg per 100 mL water). This second reason requires some careful discussion. As demonstrated by Porter and Gilfillan [26], DNC complexed to HDP (from NCZ) is more readily absorbed by chickens than the pure compound (probably as an amorphous solid). Furthermore, the authors concluded that the DNC-HDP complex dissociates through the passage in the gut. After the complex breakdown, Rogers et al. [27] suggested that DNC molecules establish intermolecular bonds, resulting in the compound precipitation as fine crystals inside the poultry gut. According to them, this solid has a greater surface area than the amorphous powder, explaining the better absorption of DNC generated from the NCZ over the directly administered compound. Thus, it is clear that litter contains the uncomplexed DNC but it does not mean lower absorption potential, because it probably occurs in the form of crystals that precipitate in the intestine. Even in the form that enhances absorption, the DNC bioavailability itself is extremely low. Again, it is possible to infer that only µg-order amounts are available for uptake through DNC ingestion at mg/kg [27].

In commercial poultry production, usually broilers are slaughtered at 42 days. In this study, breast fillet and liver were sampled also at previous ages (22 and 32 days) to monitor DNC residue in the tissues. However, for food safety purposes, the occurrence of residues only at 42 days matters, once the 10-days-withdrawal period was followed. In case of earlier slaughtering, the same withdrawal period should be considered. 

Regarding DNC residue in chicken tissues, this study considered *Codex Alimentarius* MRL (200 µg/kg) for both breast fillet and liver samples in order to verify the compliance. It is worth mentioning that EU [9,28,29] legislation is more permissive (liver = 15,000 µg/kg and chicken breast = 4000 µg/kg), whose values were recently adopted by Brazilian legislation [30] for all chicken tissues. It is interesting to notice that our non-compliant liver samples (3.7% out of 192), at 42 days, which considered the former Brazilian legislation, would now be compliant with the current legislation.

With respect to chicken litter, for all treatments at flock 1 (bedded on clean wood shavings), the DNC content showed a justifiable profile. First, it includes a rising stage resulting from the constant DNC release through broiler excreta. Then, a fall during the withdrawal period in response to a dilution effect [13], as the litter keeps gaining mass by incorporating droppings increasingly poorer in DNC. A profile similar to the first flock was seen for the other flocks (4, 7, 10) but occurring at successively ascending levels. Overall, changes in DNC concentration (within- and between flocks) have escaped initial expectations. Besides not verifying recurrence of the within-flock variations, we observed a break in longitudinal enrichment, corroborating the findings elsewhere [14]. 

Because this study has an observational purpose, we encountered limitations for explaining DNC behavior in the litter. These variations in DNC content depend, evidently, on several simultaneous effects. For instance, the input and dilution mentioned earlier are related to the volume of clean wood shavings at the first flock, housing density, dosage, withdrawal period, mortality, feed consumption and spillage of ration. In addition, evidence suggests that another issue that might influence the DNC concentration is a possible action of degrading micro-organisms, as the litter constitutes a microenvironment with high biological complexity. Unfortunately, to our knowledge, no evidence of this particular matter is available in the literature. However, a few authors demonstrated the biodegradation of DNC-like diphenylureas [31,32].

Whatever the bedding substrate for housing, broilers naturally manifest their behavior to pick at the litter. Thus, the concern on litter reuse originates from the fact that DNC persists in poultry litter. We found values (on dry basis) between 4.62–75.18 mg/kg in T1, 7.84–103.82 mg/kg in T2 and 1.96–48.30 in T3. Other studies have reported ranges of 35–152 mg/kg (on dry basis; 25 litter samples obtained from commercial poultry houses) [33], 70–90 mg/kg (basis not informed; litter used over a single flock) [12], and 5–40 mg/kg (basis not informed; litter used over a single flock) [13], and 4–1240 (on dry basis; litter used over 3 consecutive flocks) [14]. DNC levels found herein in chicken litter do not imply in residue deposition in target tissue. Additionally, Penz et al. [14] when evaluating three flocks found no detectable DNC residue in breast and drumstick from broilers raised on litter containing DNC on average between 23 and 483 mg/kg (on dry basis). 

Additionally, our pre-slaughter fasting trial did not detect DNC in breast fillets and livers after fasting for up to 12 h on contaminated reused litter. While remaining without access to feed, broilers consume litter and, at the same time, eliminate excreta. Therefore, this pre-slaughter period offers a proper time for DNC elimination that may remain circulating inside birds. Metabolic studies have demonstrated significant drops of DNC concentration in the blood plasma (45–68%) [26], meat (~32–68%) [34,35] and liver (~53%) [34] of chickens at the first 24 h after NCZ withdrawal. Altogether, these findings highlight that litter reuse does not pose risk to food safety when it comes to DNC residue. 

## 5. Conclusions

This study indicates a neglectable risk for chicken meat contamination with DNC infringing the MRL (200 µg/kg) when rearing NCZ-fed broilers in a litter reuse system. This outcome has a relevant implication for attributing safety especially to breast fillets, a world widely consumed chicken part. Finally, it is reasonable to infer that NCZ-feeding programs may be applied for broilers raised on reused litter without compromising food safety, once the additive is used obeying the dosage and withdrawal time.

## Figures and Tables

**Figure 1 animals-12-03107-f001:**
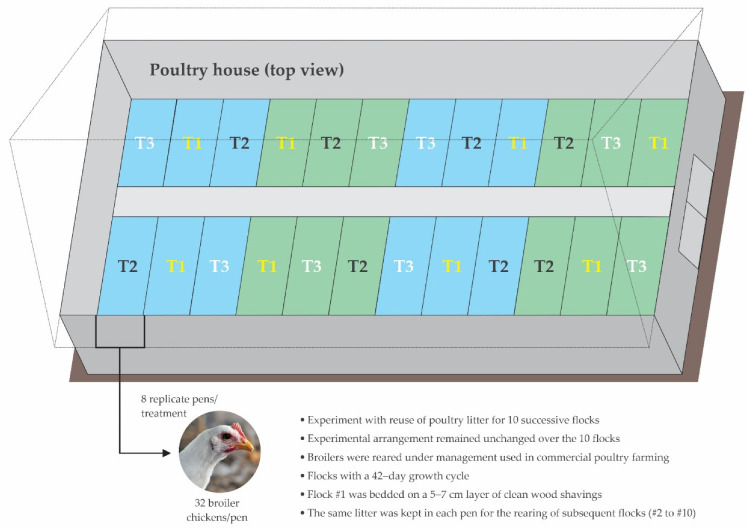
Schematic illustration of experimental setup.

**Figure 2 animals-12-03107-f002:**
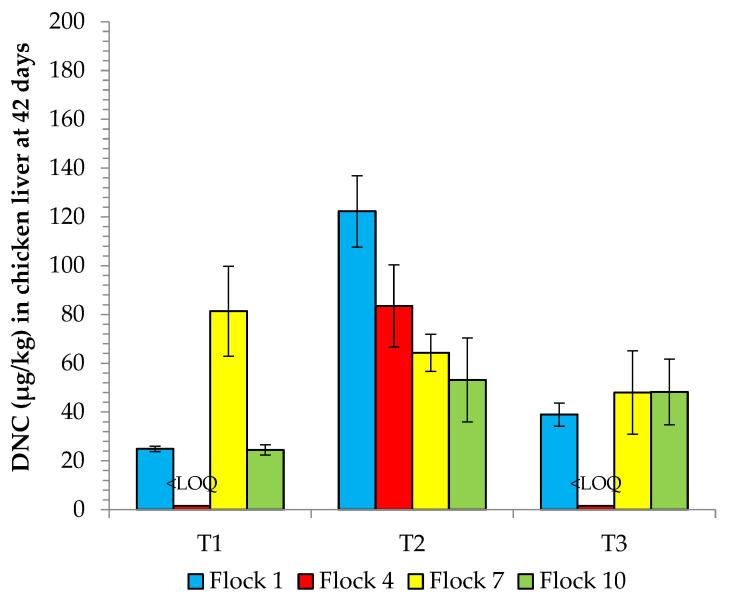
DNC residues in chicken liver at 42 days for all treatments (T1-T3). T1: NCZ at 125 mg/kg fed from 1 to 21 d; T2: NCZ at 125 mg/kg fed from 1 to 32 d; T3: NCZ at 40 mg/kg plus maduramicin at 3.75 mg/kg fed from 1 to 32 d; NCZ-free feed (withdrawal period): 22 to 42 d for T1; and 33 to 42 d for T2 and T3.

**Figure 3 animals-12-03107-f003:**
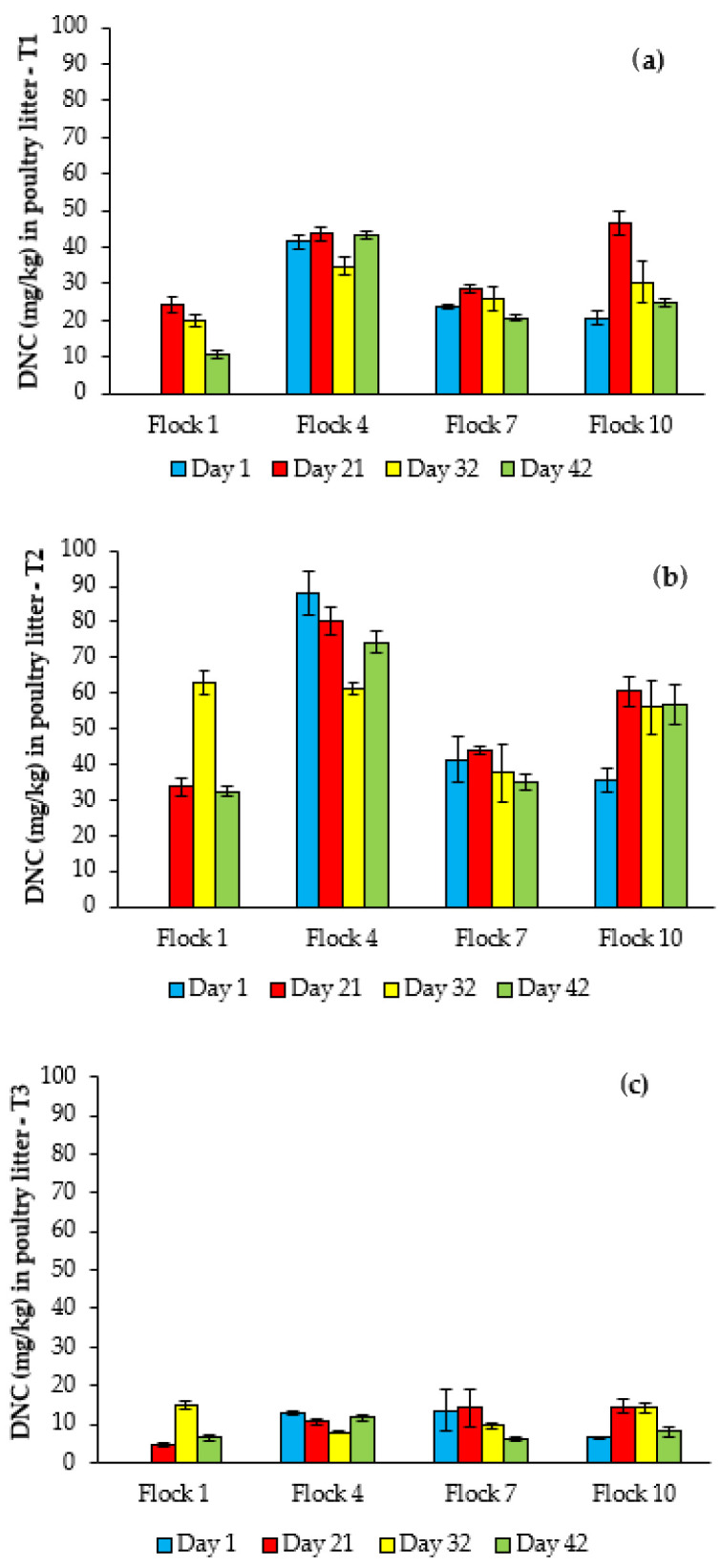
DNC concentration in poultry litter used for rearing multiple flocks of NCZ-fed broilers at 1, 21, 32 and 42 days of age in Treatment 1 (T1) (**a**), Treatment 2 (T) (**b**), and Treatment 3 (T3) (**c**). T1: NCZ at 125 mg/kg fed from 1 to 21 d; T2: NCZ at 125 mg/kg fed from 1 to 32 d; T3: NCZ at 40 mg/kg plus maduramicin at 3.75 mg/kg fed from 1 to 32 d; NCZ-free feed (withdrawal period): 22 to 42 d for T1; and 33 to 42 d for T2 and T3.

**Table 1 animals-12-03107-t001:** DNC concentration in chicken breast fillet from NCZ-fed broilers raised on poultry litter used for multiple flocks.

Treatment ^1^	Age at Slaughter (d)	DNC Concentration (µg/kg on Wet Basis) ^2,3^				*Pr > F*
		Flock 1	Flock 4	Flock 7	Flock 10	
T1 (NCZ at 125 mg/kg fed from 1–21d)	21	843.2 ± 30.0 ^ab^	926.6 ± 81.8 ^a^	644.8 ± 42.1 ^c^	707.5 ± 35.6 ^bc^	<0.0001
	32	<LOQ	<LOQ	<LOQ	<LOQ	
	42	<LOQ	<LOQ	<LOQ	<LOQ	
T2 (NCZ at 125 mg/kg fed from 1–32d)	21	783.1 ± 69.1 ^abA^	899.8 ± 55.8 ^aA^	541.2 ± 20.9 ^cA^	684.5 ± 51.3 ^bA^	<0.0001
	32	666.9 ± 77.8 ^aA^	631.8 ± 23.2 ^aB^	232.3 ± 13.8 ^cB^	474.2 ± 12.4 ^bB^	<0.0001
	42	<LOQ	<LOQ	<LOQ	<LOQ	
	*Pr > F*		<0.0001	<0.0001	<0.0001	-
T3 (NCZ at 40 mg/kg fed from 1–32d)	21	427.8 ± 27.2 ^aA^	322.2 ± 11.4 ^aA^	194.2 ± 4.5 ^bA^	275.4 ± 15.5 ^abA^	0.0158
	32	188.5 ± 23.7 ^aB^	186.7 ± 19.9 ^bB^	52.01 ± 5.47 ^bB^	124.9 ± 19.9 ^aB^	<0.0001
	42	<LOQ	<LOQ	<LOQ	<LOQ	
	*Pr > F*	<0.0001	<0.0001	<0.0001	<0.0001	-

^1^ T1: NCZ at 125 mg/kg fed from 1 to 21 d; T2: NCZ at 125 mg/kg fed from 1 to 32 d; T3: NCZ at 40 mg/kg plus maduramicin at 3.75 mg/kg fed from 1 to 32 d; NCZ-free feed (withdrawal period): 22 to 42 d for T1; and 33 to 42 d for T2 and T3. ^2^ Distinct lowercase letters in the same row differ significantly by the protected *t*-test (*p* ≤ 0.05). ^3^ Within each treatment, distinct uppercase letters in the same column differ significantly by the protected *t*-test (*p* ≤ 0.05). Values are expressed as the average of eight replicates followed by the standard-error. LOQ = 20 µk/kg.

## Data Availability

Not applicable.

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
