# Peer review of "Nicarbazin Residue in Tissues from Broilers Reared on Reused Litter Conditions"

_animals, 2022, doi:10.3390/ani12223107_

Round 1

Reviewer 1 Report

Study aimed to assess the risk of chicken meat contamination with nicarbazin used against coccidiosis, an enteric disease that affects broiler chickens. Nicarbazin was determined in liver, chicken breast, feed, and litter using a UHPLC and LC-MS/MS techniques.

According to the reviewer this Article is well prepared, method is well described, text contain essential parts, the results and discussion part is well prepared and summarized with relevant conclusions. Reviewer has minor comments and after corrections manuscript can be published in Animals.

·      The novelty of research should be clearly highlighted.

·         The Part 2.2 Experimental design is incomprehensible should be corrected and the experience should be clearly described.

·         Authors wrote that the basal diet is provided in Table S1 in Supporting Information but there are validation parameters.

·         In Tables 2, 3, 4 columns Treatment the descriptions T1, T2, T3 mislead the reader, it is explained below the table, but it should be clearly described in the table. T1 refers to the 21st day of slaughter, T2 to 32nd day of slaughter, etc.

·         Why DNC in feed and litter samples were analyzed by UHPLC, and in breast and liver by LC/MS-MS?

·         There is no chapter on method validation in the studies. There is a table with the results, but there is no information on how it was carried out and what parameters were determined.

·         The most recent literature should be cited in manuscripts, in this article 30% of the cited literature is over 10 years old.

Author Response

Dear Reviewer 1,

Thank you for your comments and attention. It helped us a lot to improve our manuscript.

Please find the answers for your comments below:

  • The novelty of research should be clearly highlighted.

    The novelty was highlighted in the last paragraph of the Introduction section.

    • The Part 2.2 Experimental design is incomprehensible should be corrected and the experience should be clearly described.

    The text was improved and a Figure (S1) was added in the Supplementary material to clarify the whole experiment.

    • Authors wrote that the basal diet is provided in Table S1 in Supporting Information but there are validation parameters.

    The tables were inverted (S1 was S2) in the Supplemental Material. The order was updated now. Thank you for the observation.

    • In Tables 2, 3, 4 columns Treatment the descriptions T1, T2, T3 mislead the reader, it is explained below the table, but it should be clearly described in the table. T1 refers to the 21st day of slaughter, T2 to 32nd day of slaughter, etc.

    We improved the description inside the former Tables 2,3,4. However we transformed some tables in Figures 1 and 2. We decided, as recommended by the other reviewer, to use more Figures along the manuscript, but even so we placed the former Tables in the Supplementary material, just in case some reader might be interested about the numeric values.

    • Why DNC in feed and litter samples were analyzed by UHPLC, and in breast and liver by LC/MS-MS?

                For chicken and liver tissues, DNC occurrence happens in trace amounts (ppb or µg/kg), requiring LC-MS/MS analysis. On the other hand, for feed and litter samples, is possible to determine NCZ in higher concentration levels (ppm or mg/kg), savings resources.

    • There is no chapter on method validation in the studies. There is a table with the results, but there is no information on how it was carried out and what parameters were determined.

                The validation methods were previously published by our group in another manuscript more devoted to chemistry studies as follows: BACILA, D. M. et al. Degradation of 4,4′-Dinitrocarbanilide in Chicken Breast by Thermal Processing. Journal of Agricultural and Food Chemistry, v. 66, n. 31, p. 8391–8397, 2018. This publication along with other two publications that involve validation of the methods used herein are reported in the last page of the Supplementary Material, however we added the References in the main document too. We followed the methodological procedures described there using control samples.

    • The most recent literature should be cited in manuscripts, in this article 30% of the cited literature is over 10 years old.

                    Regarding litter reuse and its relation with anticoccidial residues, there is indeed, no new publication. We cited Penz et al. (1999), ref.  14, however they studied only litter reuse for 3 flocks. We added some recent publications in discussion section regarding nicarbazin legislation, which was recently (2022), updated in Brazil (ref. 30), besides other two publications from 2022 (ref. 28 and 29).

Reviewer 2 Report

The manuscript describes a scientific study to analyze DNC residues in chicken meat and liver due to the intensive use of nicarbzin in poultry flocks. The study has also evaluated the possibility of reusing the litter regarding to DNC residues in chicken tissues.

The study seems interesting. It was well designed and has a good sampling. However, it is difficult to fully understand the sequence of the experiments and the main findings. So the whole text and main tables need to be better prepared before the acceptance. Some specific recommendations:

1)         To include the main results in the Abstract.

2)         To improve the Introduction, presenting a better scientific / technical background for the whole study.

3)         To explain better and clearly the sequence of the experiments in the Results. The description of some results were included only in the Discussion and they could be transferred to the Results sections.

4)         To use more figures (graphs) instead Tables in the Results. 

5)         To Improve the discussion after making the other changes.

I also suggest the authors to reconsider the title proposed. I think it should inform it was a field study. 

Author Response

Dear Reviewer,

Thank you for your comments and attention. It helped us a lot to improve our manuscript.

1)         To include the main results in the Abstract.

We added four lines in the Abstract about the results.

2)         To improve the Introduction, presenting a better scientific / technical background for the whole study.

We improved the explanation why the study was done.

3)         To explain better and clearly the sequence of the experiments in the Results. The description of some results were included only in the Discussion and they could be transferred to the Results sections.

We improved the experimental section and included one Figure (S1) to elucidate the sequence order of the experiment in the Supplementary material. The results were improved: some Tables were replaced by Figures in the main document. Although we left the Tables in the Supplementary material, if someone would be interested in more details.

4)         To use more figures (graphs) instead Tables in the Results. 

We improved the description inside the Tables, although we moved some of them into the Supplementary material section. We transformed some tables in Figures 1 and 2. 

5)         To Improve the discussion after making the other changes.

We improved the discussion. Thank you.

I also suggest the authors to reconsider the title proposed. I think it should inform it was a field study. 

We changed the title a little bit. Although the experiment was done in an experimental broiler house, which resembles a field study,  it can not be considered a commercial field farm experiment.

Considering the complexity of our research, the experiment would be unfeasible in real field conditions. Some of the reasons would be the difficulty to access the farm due to biosecurity issues, difficulty to use different diets (treatments), because the feeder is unique for the whole aviary and the duration of the experiment (2 years). Besides, the control of chicken litter collection from different treatments would be unfeasible.

Round 2

Reviewer 2 Report

The manuscript has really been improved over the first version. The study is clearer and better described now. The title is also fitting more with the article now. It was also important to prepare and attach the Supplementary Material for a better understanding of the experiments. However, the results are still difficult to understand. So I have two main recommendations for authors:

1) The Summary was really improved after including some Results (it was necessary!). But I don't think they should have been included in the Simple Summary. The phrase “All samples collected from broilers at 42 days of age (at slaughter) showed concentration below the maximum residue levels (200 µg/kg)” could be removed.  

2) Results section titles are not reporting the specific findings described. It is also relatively difficult to understand the main findings. Therefore, I recommend an additional effort by the authors to change the section titles and better describe the main results in the topics. Regarding the sequential order of subjects, I think that Table 1 should be presented in the last section of the Results. Please also avoid a one-sentence section (3.2). This sentence should be included in another section.

Also the entire text needs an additional proofreading to correct some few mistakes (just one example: “4 females and 4 females”  in the lines 119-120) and to use more appropriate scientific sentences (please avoid “as expected”, “contrary to expectations”, etc.).

Author Response

The manuscript has really been improved over the first version. The study is clearer and better described now. The title is also fitting more with the article now. It was also important to prepare and attach the Supplementary Material for a better understanding of the experiments. However, the results are still difficult to understand. So I have two main recommendations for authors:

Thank you.

1) The Summary was really improved after including some Results (it was necessary!). But I don't think they should have been included in the Simple Summary. The phrase “All samples collected from broilers at 42 days of age (at slaughter) showed concentration below the maximum residue levels (200 µg/kg)” could be removed. 

Amended.

2) Results section titles are not reporting the specific findings described. It is also relatively difficult to understand the main findings. Therefore, I recommend an additional effort by the authors to change the section titles and better describe the main results in the topics. Regarding the sequential order of subjects, I think that Table 1 should be presented in the last section of the Results. Please also avoid a one-sentence section (3.2). This sentence should be included in another section.

Amended. The main finding was included in the first paragraph of the Result section. We deleted the title in item 3.2 and included its sentences inside Discussion section.

We discussed this issue among the team and we think Table 1 can be placed in the Supplementary material (SM) section. This table is important for the further discussion of the results. As we dealt with what is significant or not in the text itself, it is not so relevant keep it within the main document. In order to balance the amount of Figures and Tables placed in the main document and in the SM, we moved Figure S1 to Figure 1 (main document).

Also the entire text needs an additional proofreading to correct some few mistakes (just one example: “4 females and 4 females” in the lines 119-120) and to use more appropriate scientific sentences (please avoid “as expected”, “contrary to expectations”, etc.).

Amended. The suggestion was accepted. The text was corrected along with other minor mistakes along the text.